# Effects of Elastic Band Training on Physical Performance in Team Sports: A Systematic Review and Meta-Analysis

**DOI:** 10.3390/jfmk10040402

**Published:** 2025-10-17

**Authors:** Dušan Stanković, Anja Lazić, Nebojša Trajković, Miladin Okičić, Aleksa Bubanj, Tomáš Vencúrik, Tomislav Gašić, Saša Bubanj

**Affiliations:** 1Faculty of Sport and Physical Education, University of Niš, 18000 Niš, Serbia; dukislavujac@gmail.com (D.S.); anja.lazic96@hotmail.com (A.L.); nele_trajce@yahoo.com (N.T.); miladinokicic.2001@gmail.com (M.O.);; 2Faculty of Medicine, University of Niš, 18000 Niš, Serbia; bubanjaleksa@gmail.com; 3Department of Sports, Faculty of Sports Studies, Masaryk University, 62500 Brno, Czech Republic; vencurik@fsps.muni.cz; 4Department of Sports Games, Faculty of Physical Education and Sport, Comenius University Bratislava, 814 99 Bratislava, Slovakia

**Keywords:** resistance training, resistance band, strength

## Abstract

**Objectives:** Elastic band training is a popular alternative to traditional resistance methods, but its effects on sport-specific performance in team athletes remain inconsistent. This systematic review and meta-analysis aim to evaluate the efficacy of elastic band training on muscular strength, linear sprint, change of direction (COD), and jump height in team sport athletes. **Methods:** Following PRISMA guidelines, a comprehensive search was conducted in PubMed, Web of Science, and Scopus for randomized controlled trials and quasi-experimental studies. The quantitative synthesis included studies comparing elastic band training interventions with control groups receiving routine training, habitual physical activity, or no additional resistance training intervention. Data were extracted using a standardized form, and a meta-analysis was performed using a random-effects model. Standardized mean differences (SMDs) and 95% confidence intervals (CIs) were calculated to determine the pooled effect of the intervention on key performance indicators. A total of 729 athletes were included. **Results:** The meta-analysis showed a statistically significant positive effect of elastic band training on lower limb explosive power (SMD = 1.43, *p* = 0.01), change of direction performance (SMD = −2.54, *p* = 0.01), and sprint performance (SMD = −1.64, *p* = 0.01). **Conclusions:** Elastic band training is a highly effective and practical method for significantly improving key physical performance indicators, including explosive power, COD, and sprint ability, in team sport athletes. Its portability and adaptability make it a valuable alternative or complement to conventional resistance training.

## 1. Introduction

Physical performance in team sports is a critical factor influencing an athlete’s ability to execute sport-specific movements, such as sprinting, jumping, change of direction (COD), and strength-based actions [1]. Strength training is widely recognized as a fundamental component of athletic development, with various modalities employed to enhance performance and reduce injury risk [2]. Traditional resistance training methods, such as free weights and machines, have been extensively studied; however, alternative approaches, including elastic band training, have gained increased attention due to their practicality, adaptability, and ability to provide variable resistance throughout the range of motion [3].

Elastic band training has been incorporated into strength and conditioning programs across different sports, offering an effective means of improving muscular strength, power, and endurance [4]. Previous research has suggested that elastic resistance training can elicit comparable neuromuscular adaptations to conventional resistance training methods [5]. Nevertheless, the effectiveness of elastic band exercises on sport-specific physical performance parameters in team sports remains to be systematically evaluated [6].

Elastic band training offers a unique biomechanical stimulus compared to conventional fixed-load resistance training, providing variable resistance where an increase in tension is directly proportional to the elongation of the band. This specific load profile enables the resistance to accommodate the natural muscle strength curve. Consequently, the athlete is maximally challenged at the end of the concentric range of motion, which is often underloaded with free weights [6]. This unique profile is hypothesized to specifically influence factors critical for athletic performance, such as rate of force development [7]. Furthermore, some evidence suggests that the varying resistance may promote greater muscle activation [8]. Understanding these underlying neural and mechanical adaptations is crucial for justifying the use of elastic band training. Despite the increasing use of elastic bands in athletic training, evidence regarding their efficacy in enhancing physical performance remains inconsistent [9]. Some studies report significant improvements in sprint speed, COD, and jump height, while others indicate minimal or negligible effects [10]. Moreover, variations in training protocols, athlete characteristics, and outcome measures contribute to the heterogeneity of findings [11]. Therefore, a systematic review and meta-analysis are necessary to synthesize existing evidence and provide a comprehensive assessment of the impact of elastic band training on physical performance in team sports.

The primary objective of this systematic review and meta-analysis is to evaluate the effects of elastic band training on key physical performance indicators in team sport athletes. Specifically, this study aims to determine the extent to which elastic resistance training influences strength, power, linear sprint performance, COD, and jump height in team sport contexts. By systematically analyzing the available literature, this review will provide evidence-based recommendations for coaches, practitioners, and researchers regarding the utility of elastic band exercises in athletic performance enhancement.

## 2. Materials and Methods

This systematic review and meta-analysis were conducted in accordance with the Preferred Reporting Items for Systematic Reviews and Meta-Analyses (PRISMA) guidelines [12]. As required by the guidelines for methodologically sound drafting of systematic reviews, the protocol of this systematic review was registered with the International Platform of Registered Systematic Review and Meta-Analysis Protocols (INPLASY, registration number: INPLASY202590067). The review includes studies that examine the effects of elastic band training on physical performance measures in team sport athletes.

### 2.1. Eligibility Criteria

According to the PICO (Population, Intervention, Comparison, and Outcome) strategy [13], our research was characterized as follows: Population: Studies including male and female athletes competing in team sports (e.g., basketball, soccer, volleyball, handball, rugby) at various competitive levels; Intervention: Training interventions utilizing elastic bands as a primary resistance modality for improving physical performance; Comparison: Control groups not performing traditional resistance training, or other strength and conditioning programs; Outcomes: Measures of physical performance, including muscular strength, power, sprint speed, COD, and jump height; Study Design: Randomized controlled trials (RCTs), quasi-experimental studies, and cohort studies published in peer-reviewed journals.

### 2.2. Search Strategy

A comprehensive literature search was conducted across multiple electronic databases, including PubMed, Web of Science, and Scopus. The search strategy will combine relevant keywords and Boolean operators, such as ((“(Resistance Training”[MeSH]) OR (“elastic band”[tiab]) OR (“resistance band”[tiab]) OR (“rubber band”[tiab]) OR (“Thera-Band”[tiab]) OR (“resistance tubing”[tiab])) AND ((“Athletes”[MeSH]) OR (“team sport”[tiab]) OR (“soccer”[tiab]) OR (“basketball”[tiab]) OR (“volleyball”[tiab]) OR (“handball”[tiab]) OR (“rugby”[tiab])) AND ((“Athletic Performance”[MeSH]) OR (“strength”[tiab]) OR (“power”[tiab]) OR (“jump performance”[tiab]) OR (“sprint speed”[tiab]) OR (“agility”[tiab])). Additional manual searches were performed using reference lists of included articles to identify any relevant studies not captured in the initial search. The electronic database search included studies published between January 2014 and September 2025.

### 2.3. Study Selection and Data Extraction

The selection process involved the following steps: (i) Reviewer A searched the database for relevant articles and imported them into Rayyan web-based tool for literature screening. (ii) The same reviewer screened the records, removing duplicates, systematic reviews, meta-analyses, and non-English articles. (iii) Two independent reviewers (A and B) reviewed the titles and abstracts. Any discrepancies between reviewers were resolved through discussion or consultation with a third reviewer (C). (iv) One reviewer extracted the data, and another verified its accuracy. (v) Two independent reviewers assessed the methodological quality of the included studies. Full-text articles meeting the inclusion criteria were further reviewed for final inclusion. All data (scientific papers) were extracted and imported into the “Rayyan” web-based tool for systematic literature review. Additionally, the “Blind ON” option was utilized to reduce bias and ensure independent screening during the selection process of scientific papers. WebPlotDigitizer online software (Version 4.8) was used to collect data when results in studies were presented graphically.

Data was extracted using a standardized form, capturing the following information: study characteristics (authors, year, country); participant characteristics (sample size, age, sex, sport, and competitive level); intervention details (training duration, frequency, intensity, and exercise selection); outcome measures and key findings; and risk of bias assessment using the Cochrane Risk of Bias Tool for randomized studies.

### 2.4. Statistical Analysis

A meta-analysis was conducted using MedCalc statistical software version 19.6 (MedCalc Software Ltd., Ostend, Belgium). Standardized mean differences (SMDs) and 95% confidence intervals (CIs) were calculated for continuous outcome variables. Effect size was measured using the standardized mean difference (SMD). The pooled SMD was calculated with a *p*-value of 0.05. According to Cohen’s, an SMD of 0.2 is considered small, 0.5 is medium, and 0.8 is large [14]. A random-effects model was used to account for variability across studies. Heterogeneity was assessed using the I^2^ statistic, with values categorized as low (<25%), moderate (25–50%), and high (>50%). Sensitivity analyses were performed to assess the robustness of findings, and publication bias was evaluated using funnel plots and Egger’s regression test.

## 3. Results

A database search yielded 1612 articles. After removing 677 duplicates, the remaining 935 articles were screened by title and abstract, which led to the exclusion of 869 publications. A total of 66 articles underwent full-text assessment, and 45 were excluded for not meeting the inclusion criteria. This resulted in a final count of 21 studies included in the analysis. The full process is shown in the flow diagram (Figure 1).

### 3.1. Quality Assessment

#### 3.1.1. Publication Bias and Heterogeneity

The revised Cochrane (RoB2) tool for assessing the risk of bias in randomized studies was used to evaluate potential bias risks [15]. This tool covers five domains of bias risk: (1) randomization process; (2) deviations from intended interventions; (3) missing outcome data; (4) outcome measurement methods; (5) selection of reported results. An overall risk of bias assessment was determined for each study. One researcher conducted the bias risk assessment, and in cases where evaluation was challenging, advice from a second researcher was sought. All studies exhibited a low risk of bias in the third domain (missing outcome data). A detailed analysis is presented in Figure 2. The risk of bias summary for non-randomized studies is shown in Figure 3.

#### 3.1.2. Characteristics of the Sample

The analyzed studies included participants from ten different countries (Indonesia, France, Brazil, Lithuania, Spain, Denmark, Tunisia, Germany, Norway, and Turkey). In twelve studies, the participants were handball players [3,9,16,17,18,19,20,21,22,23,24,25]. Football players were included in four studies [10,26,27,28]. Two studies included volleyball players [29,30], while one study each examined basketball players [31], rugby players [32], and kabaddi players [33]. The total sample across all studies included in this systematic literature review consisted of 729 participants. The smallest and largest sample sizes were 12 [3] and 319 [28], respectively. Male participants were the focus of thirteen studies [9,10,16,17,18,19,20,26,27,30,31,32,33]. All studies included one experimental and one control group, except for a single study that involved four experimental groups [33].

**Figure 2 jfmk-10-00402-f002:**
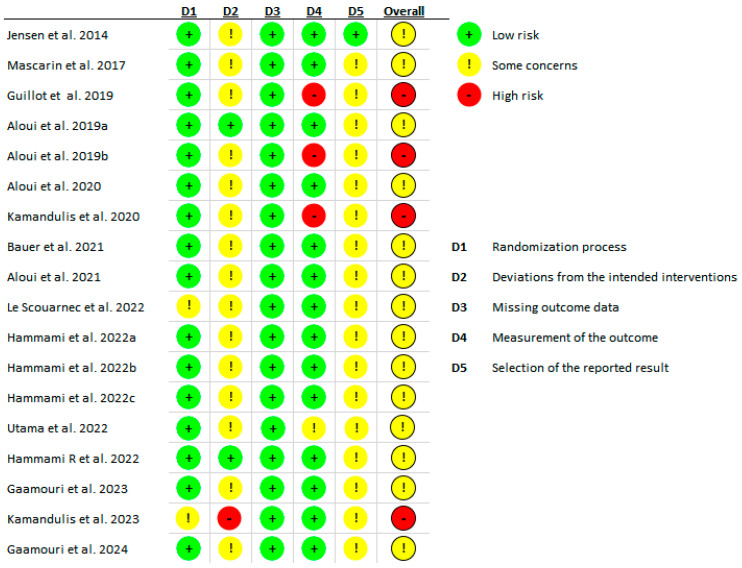
Risk of bias summary of included studies [9,16,17,18,19,20,21,22,23,24,25,26,27,28,30,31,32,33].

**Figure 3 jfmk-10-00402-f003:**
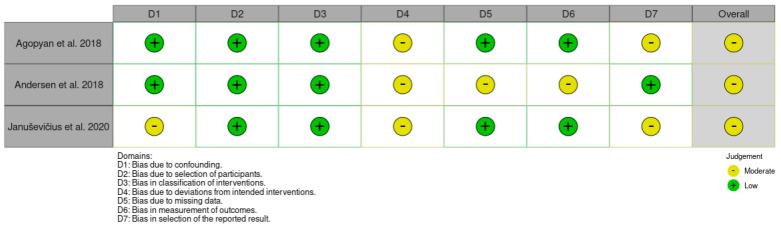
Risk of bias summary of non-randomized studies [3,10,29].

#### 3.1.3. Characteristics of Strength Training Programs

The duration of interventions ranged from 5 to 10 weeks, with two to three sessions per week. The most common protocol involved two sessions per week for 8 weeks [16,17,18,20,26,27,29,30]. Eight studies investigated the effects of elastic resistance training on vertical jump performance [10,16,18,21,22,23,24,29], and another eight focused on sprint performance [10,16,18,21,22,23,24,27]. Four studies assessed balance [21,22,23,30]. Eight studies examined upper body strength [9,19,20,21,22,23,24,29], and seven studies analyzed lower body strength following resistance training with elastic bands [10,16,18,24,26,27,30]. Many studies used more than eight repetitions per exercise during elastic band training [9,16,17,19,21,22,23,24,25,26,27,29,30]. Two studies [18,20] employed a lower number of repetitions in plyometric training with absolute load. Three studies used a maximum number of repetitions within a time interval of 4 s [10,28,31]. Four studies estimated the maximum load with elastic bands using the subjective perception of effort method [3,9,26,29]. A total of 13 studies applied absolute load values in elastic resistance training [16,17,18,19,20,21,22,23,24,25,27,30,33]. Using the maximum number of repetitions within a given time period was reported in three studies [10,28,31]. Detailed characteristics of included studies were shown in Table 1.

#### 3.1.4. The Effect of Resistance Band Strength Training on Lower Extremities

The effect size of strength training with elastic resistance bands (pre- and post-intervention) on lower limb explosive power demonstrated a statistically significant pooled standardized mean difference (SMD = 1.43; *p* = 0.01) (Figure 4).

#### 3.1.5. The Effect of Resistance Band Strength Training (Pre- and Post-Intervention) on COD Performance

The meta-analysis of the effect of resistance band strength training on COD performance revealed a statistically significant pooled standardized mean difference (SMD = −2.54; *p* = 0.01) (Figure 5).

#### 3.1.6. Meta-Analysis Results Indicate a Significant Positive Effect of Resistance Band Strength Training on Speed Performance (Sprint) Pre- and Post-Intervention

The meta-analysis results showed a statistically significant positive effect of resistance band strength training on sprint performance, with a pooled standardized mean difference (SMD = −1.64; *p* = 0.01). The negative SMD reflects improved sprint performance, as faster sprint times indicate better results (Figure 6).

## 4. Discussion

This meta-analysis aimed to comprehensively evaluate the effects of elastic resistance band training on key physical performance attributes in team sport athletes, specifically lower-limb power, COD, and sprint performance. The findings demonstrate that resistance band training elicits statistically significant and practically meaningful improvements in all three performance domains.

Lower limb explosive power is the capacity to produce high force at high velocity, making it a critical component for success in various competitive sports [34]. In our study, the pooled effect size for lower-limb explosive power (SMD = 1.43, *p* = 0.01) indicates a large positive effect, supporting the efficacy of elastic resistance training in enhancing vertical jump and superior adaptive responses in trained athletes. The meta-analysis by de Oliviera et al. [35] on elastic band training was not superior to active controls for improving lower limb muscle strength in healthy, non-athletic participants. It is important to say that this study was limited to only a few studies included in the meta-analysis, as well as methodological heterogeneity [35]. These specific benefits are likely due to the variable loading profile of the elastic bands, which is also supported in comparative studies. Variable resistance training (elastic band combined with weights) is more effective in enhancing lower limb explosive power than constant-load training [36]. This aligns with prior research suggesting that elastic bands effectively increase force throughout the concentric phase, effectively increasing neuromuscular activation in multi-joint lower-body exercises [3,33], contributing to adaptations relevant for explosive performance in sports such as handball, football, and basketball.

The ability to change direction effectively depends largely on an athlete’s physical attributes, such as eccentric braking capabilities to halt momentum before accelerating in a new direction [37]. Change of direction performance also showed a very large and statistically significant effect (SMD = –2.54, *p* = 0.01) after elastic band training. The negative SMD reflects reduced completion time in COD tests, indicating improved performance. The positive effects of elastic band training can be compared against other common resistance modalities. Traditional resistance training yielded medium effects on COD ability (SMD = 0.62). Furthermore, even highly advanced methods like flywheel resistance training reported large effects (SMD = 1.63) in similar athletic cohorts [38]. Change of direction is a crucial component in team sports, where rapid changes of direction are frequent. The large magnitude of improvement suggests that elastic band training may target key muscular and neuromechanical components of COD, including reactive strength and deceleration control. This can be explained by the variable resistance profile of elastic bands, which provides increasing tension during the concentric, accelerating phase, and enhances the loading demand on the musculature during the critical eccentric (braking) phase required before the change of direction. The training method using elastic bands can be especially valuable during in-season or injury-preventive training phases, where elastic bands offer a joint-friendly and adaptable resistance modality [39].

Sprint performance improved significantly as well, with a pooled SMD of –1.64 (*p* = 0.01), again reflecting better sprint times post-intervention. This is consistent with the notion that elastic resistance training enhances acceleration mechanics by increasing force output during the propulsion phase. Given the frequency and importance of short sprints in team sports, these findings provide practical support for integrating elastic band training into conditioning programs aimed at improving game-related speed.

Overall, these results confirm that elastic resistance bands are an effective modality for improving physical performance in team sport athletes. Beyond their physiological benefits, resistance bands are cost-effective, portable, and allow for multidirectional loading, making them especially attractive for use in team settings with limited equipment or space. A crucial consideration for interpreting these results is the nature of the control groups. We must clarify that all control groups consistently performed only active routine training (e.g., normal team practice) and did not receive a dedicated conventional resistance intervention. Consequently, the observed positive effect sizes should be interpreted as the efficacy of integrating elastic band training into an athlete's existing routine programming. They should not be mistaken as evidence of elastic band training's inherent superiority over other structured resistance training modalities (such as free weights at a matched intensity). Clear conclusions about head-to-head superiority remain beyond the scope of this systematic review, as it requires dedicated comparative randomized controlled trials.

However, several limitations should be acknowledged. Heterogeneity across studies was substantial, possibly due to variations in training protocols, participant age and performance level, or assessment methods. While the use of a random-effects model accounted for this variability, future research should seek to standardize intervention parameters to better isolate the mechanisms of adaptation, as well as to compare the effects of elastic band training against other common resistance training modalities on physical performance measures in team sports. A significant limitation is the inherent variation in elastic band specifications, such as size and material stiffness, which directly determines the resistance load and varies widely across manufacturers. While some studies tried to standardize the load (using elongation percentages or mass equivalents), the precise relationship between band type and effective resistance across all included studies remains unclear. Therefore, while our findings support the efficacy of elastic band training, future randomized controlled trials are needed to establish the minimum effective stimulus and the optimal dosage (frequency, duration, and load) required to maximize performance enhancement. Furthermore, most included studies had relatively short durations (5–10 weeks), so long-term effects remain unclear.

## 5. Conclusions

This meta-analysis provides strong evidence that resistance band strength training significantly improves lower limb explosive power, COD, and sprint performance in team sport athletes. The observed effect sizes indicate that elastic band training can be an effective and practical method for enhancing key performance parameters relevant to multidirectional and high-speed sports contexts.

Given its portability, affordability, and adaptability, resistance band training can serve as a viable alternative or complement to traditional resistance methods, particularly when equipment access is limited or when joint-friendly loading is required. Strength and conditioning professionals are encouraged to integrate elastic bands into periodized training programs, especially during phases that emphasize speed, COD, and power development.

## Figures and Tables

**Figure 1 jfmk-10-00402-f001:**
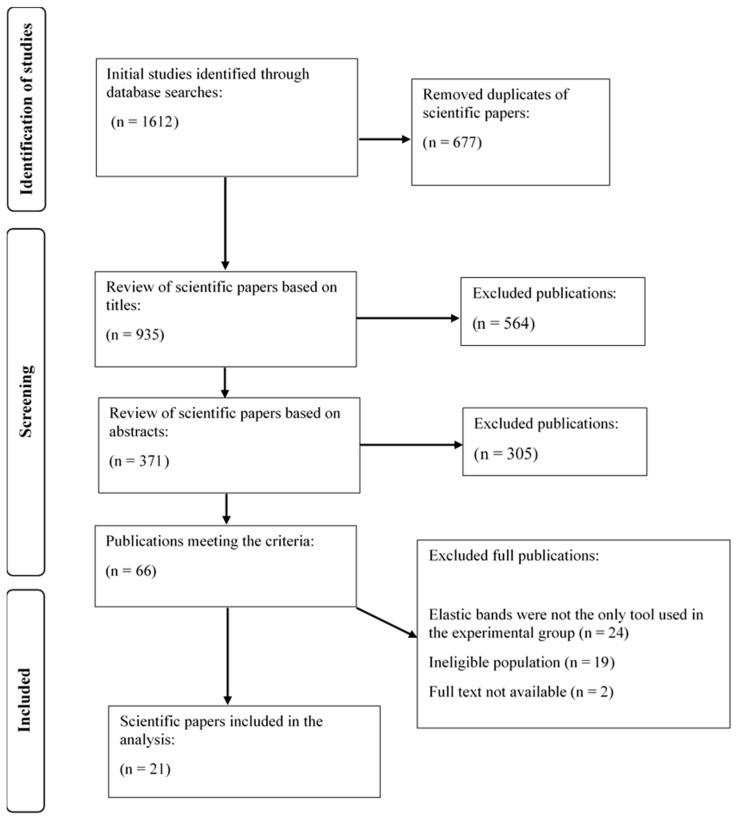
PRISMA flow diagram.

**Figure 4 jfmk-10-00402-f004:**
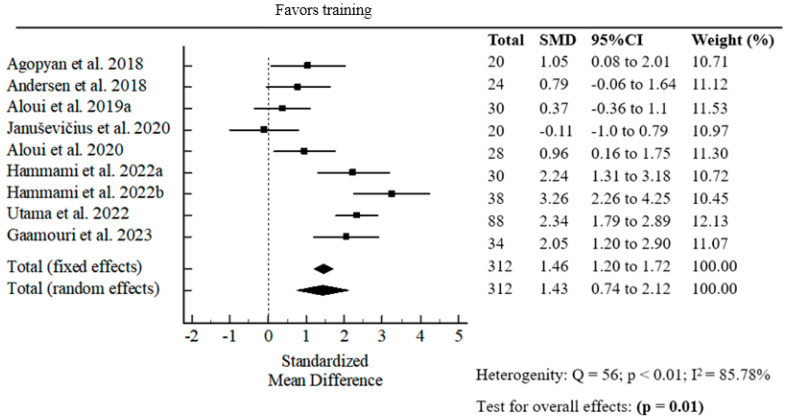
Forest plot of the pooled standardized mean difference for resistance band strength training in vertical jump performance [3,10,16,18,21,22,24,29,33].

**Figure 5 jfmk-10-00402-f005:**
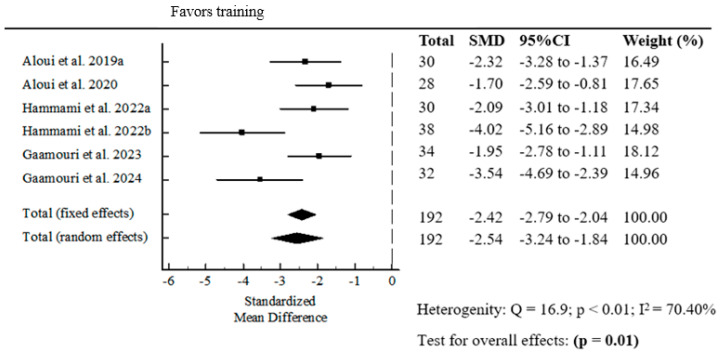
Forest plot of the pooled standardized mean difference for resistance band strength training in COD performance [16,18,21,22,24,25].

**Figure 6 jfmk-10-00402-f006:**
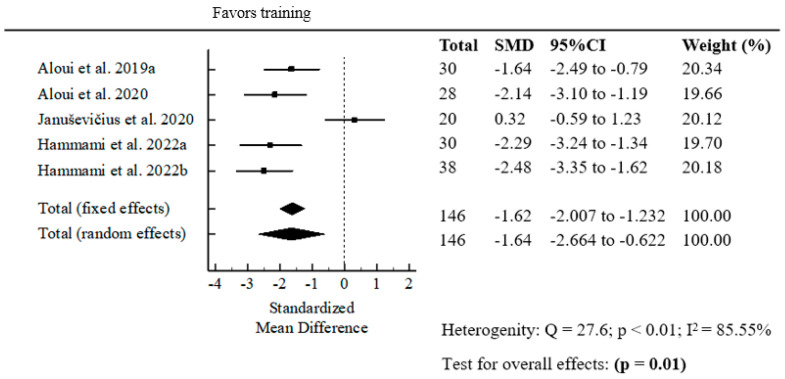
Forest plot of the pooled standardized mean difference for resistance band strength training on sprint performance [10,16,18,21,22].

**Table 1 jfmk-10-00402-t001:** Characteristics of included studies.

Authors	Sport	Participants (Number, Age, and Gender)	Tested Variables	Training Program	Repetitions per Set, Sets, and Load	Results
Jensen et al., 2014 [26]	Football	34, 22.1 ± 3.3, males	Eccentric and isometric hip adduction strength	EG: Hip adductor exercises EBT for 8 weeks (2–3 sessions/week) + routine training; CG: Routine training	3 sets 8 RM, 10 RM, 15 RM	Eccentric hip adduction strength ↑; isometric hip adduction strength ↔
Mascarin et al., 2017 [9]	Handball	39, 15.2 ± 1.0, females	Shoulder rotation strength, ball throw speed	EG: EBT for 6 weeks (3 sessions/week) + routine training; CG: Routine training	3 sets 10 RM, 15 RM, 20 RM	Shoulder rotation strength, ball throw speed ↑
Agopyan et al., 2018 [29]	Volleyball	20, 16.2 ± 0.7, females	Vertical jump and maximum strength, upper extremity speed	EG: EBT for 8 weeks (2 sessions/week) + routine training; CG: Routine training	3 sets 12 RM	Vertical jump and maximum strength ↑; upper extremity speed ↔
Andersen et al., 2018 [3]	Handball	12, 16.5 ± 0.7, females	LL, UL strength, COD, vertical jump	EBT for 9 weeks (3 sessions/week)	3 sets 6–10 RM	LL, UL strength ↑; COD ↔
Guillot et al., 2019 [32]	Rugby	23, 17.2 ± 0.6, males	Range of motion of the joint	EG: Mobility training with EB for 5 weeks (24 sessions); CG: Passive postural training LL	/	Range of motion of the joint ↑
Aloui et al., 2019 [16]	Handball	30, 18.7 ± 0.8, males	LL strength, 5 m and 30 m sprint, COD, vertical jump	EG: Part of routine training replaced with EBT for 8 weeks (2 sessions/week); CG: Routine training	3 sets 12–15 R 16–34 kg (200%)	5 m and 30 m sprint, COD, LL strength ↑; vertical jump ↔
Aloui et al., 2019 [17]	Handball	30, 18.7 ± 0.8, males	UL strength	EG: Part of routine training replaced with EBT for 8 weeks (2 sessions/week); CG: Routine training	3 sets 12–15 R 7–18 kg (250%)	UL strength ↑
Kamandulis et al., 2020 [31]	Basketball	18, 21.5 ± 1.7, males	LL movement speed, muscle strength, neurological control of posterior muscle chain, 30 m sprint	EG: Knee joint flexion with EBT for 5 weeks (3 sessions/week) + routine training; CG: Routine training	4–6 sets Maximum repetitions for 4 s 4.6 kg (100–300%)	LL movement speed, muscle strength, neurological control of posterior muscle chain, 30 m sprint ↑
Aloui et al., 2020 [18]	Handball	29, 17.9 ± 0.4, males	LL strength, muscle girth, 5 m and 30 m sprint, COD, vertical jump	EG: Part of routine training replaced with EBT and plyometrics for 8 weeks (2 sessions/week); CG: Routine training	5 sets 6 R 41.4 kg–81 kg	LL strength, 5 m and 30 m sprint, COD ↑; muscle girth, LL strength, vertical jump ↔
Januševičius et al., 2020 [10]	Football	20, 23.5 ± 7.2, males	Knee extension-flexion movement frequency, vertical jump, 30 m sprint, knee flexion, and concentric extension torque	EG: Calf flexion using EB for 5 weeks (3 sessions/week) + routine training; CG: Routine training	4–6 sets Maximum repetitions for 4 s 4.6 kg (100–300%)	Knee extension-flexion movement frequency ↑; vertical jump, 30 m sprint, knee flexion and concentric extension torque ↔
Aloui et al., 2021 [20]	Handball	29, 17.9 ± 0.4, males	UL strength	EG: Part of routine training replaced with EBT and plyometrics for 8 weeks (2 sessions/week); CG: Routine training	5 sets 6 R 11.2–26 kg	UL strength ↑
Bauer et al., 2021 [19]	Handball	32, 17.0 ± 0.8, males	Strength endurance and isometric UL strength	EG: Part of routine training replaced with EBT for 9 weeks (3 sessions/week); CG: Routine training	3 sets 8–12 R 2 kg (100%)	Strength endurance and isometric UL strength ↑
Utama et al., 2022 [33]	Kabaddi	44, 17.6 ± 1.0, males	COD	EBT for 6 weeks (3 sessions/week); G1: Low-intensity EBT with 1.77 kg load; G2: Low-intensity EBT with 2.27 kg load; G3: High-intensity EBT with 1.77 kg load; G4: High-intensity EBT with 2.27 kg load	6–8 sets 4–6 R 1.7 kg–2.27 kg	COD ↑
Hammami R et al., 2022 [30]	Volleyball	27, 14.8 ± 0.4, males	LL strength, dynamic strength, reactive strength, horizontal center of pressure displacement, movement speed, center of pressure oscillation area, anteroposterior balance	EG: EBT for 8 weeks (2 sessions/week) + Routine training; CG: Routine training	2–4 sets 10–15 R 16–34 kg (200%)	LL strength, dynamic strength, anteroposterior balance ↑; reactive strength, horizontal center of pressure displacement, movement speed, center of pressure oscillation area ↔
Le Scouarnec et al., 2022 [27]	Football	15, 17.5 ± 0.3, males	Force production in the anteroposterior direction, 30 m sprint	EG: Sprints with EBT for 8 weeks (2 sessions/week); CG: Sprints	2 sets 12–14 sprint R	Force production in the anteroposterior direction, 30 m sprint ↑
Hammami et al., 2022 [21]	Handball	30, 15.7 ± 0.3, females	UL strength, 5 m, 10 m, 20 m, and 30 m sprint, COD, vertical and horizontal jump, balance	EG: Part of routine training replaced with EBT for 10 weeks (2 sessions/week); CG: Routine training	3–5 sets 12 R 4.4 kg–16 kg (200%)	UL strength, 5 m, 10 m, 20 m, and 30 m sprint, COD, vertical jump ↑; horizontal jump, balance ↔
Hammami et al., 2022 [22]	Handball	38, 15.8 ± 0.2, females	UL strength, 5 m, 10 m, 20 m, and 30 m sprint, COD, vertical and horizontal jump, balance	EG: Part of routine training replaced with EBT for 10 weeks (2 sessions/week); CG: Routine training	3–5 sets 12 R 32 kg–16 kg (250%)	UL strength, 5 m, 10 m, 20 m, and 30 m sprint, COD, vertical jump, horizontal jump ↑; balance ↔
Hammami et al., 2022 [23]	Handball	26, 15.8 ± 0.2, females	UL strength, 5 m, 10 m, 20 m, and 30 m sprint, COD, vertical and horizontal jump, balance	EG: Part of routine training replaced with EBT for 10 weeks (2 sessions/week); CG: Routine training	3–5 sets 10 R 32 kg–16 kg (250%)	UL strength, 20 m and 30 m sprint, COD, vertical jump ↑; 5 m, 10 m sprint, horizontal jump, balance ↔
Gaamouri et al., 2023 [24]	Handball	34, 15.8 ± 0.2, females	LL and UL strength, COD, vertical and horizontal jump, repeated 20 m sprint ability	EG: Part of routine training replaced with EBT for 10 weeks (2 sessions/week); CG: Routine training	3–5 sets 10 R 32 kg–16 kg (250%)	LL and UL strength, COD, vertical jump, repeated 20 m sprint ability ↑; horizontal jump ↔
Kamandulis et al., 2023 [28]	Football	319, 23.2 ± 4.8, males	Effectiveness in preventing posterior thigh muscle injuries	EG: Knee joint flexion with EBT for 5 weeks (2–3 sessions/week) + routine training; CG: Football-specific exercises at own pace + routine training	2–6 sets 4 s RM 8.6 kg	Prevention of posterior thigh muscle injuries ↔
Gaamouri et al., 2024 [25]	Handball	30, 15.8 ± 0.2, females	Modified *t*-test, squat jump, countermovement jump, standing long jump, maximum chest press, half-squat	EG: 10 weeks (2 sessions/week) + routine training; CG: Routine training	3–5 sets 12 R 3.2 kg–8.8 kg (folded band)	Modified *t*-test, squat jump, countermovement jump, standing long jump, maximum chest press, half-squat ↑

Legend: LL—lower limbs; UL—upper limbs; EG—experimental group; CG—control group; RM—repetition maximum; R—number of repetitions; %—percentage of elastic band elongation; COD—change of direction; EBT—elastic band training; ↑—improvement; ↔—unchanged.

## Data Availability

No new data were created or analyzed in this study. Data sharing is not applicable to this article.

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
