# Peer review of "Effects of Elastic Band Training on Physical Performance in Team Sports: A Systematic Review and Meta-Analysis"

_jfmk, 2025, doi:10.3390/jfmk10040402_

Round 1

Reviewer 1 Report

Comments and Suggestions for Authors

This is a meta analysis on the effects of resistance training with elastic bands on a series of physical performance tests. In general, the methodology (steps for searching and examining the studies and analyzing the data) is appropriate. I have some comments for the authors to address:

  1. Elastic bands can come in different sizes and different stiffness. Has this been taken into consideration when evaluating the results? If not, then a comment is needed in the discussion.
  2. The variability in training programs needs specific commentary. As stated in the article, some studies have examined 2 weeks of training others 10 weeks of training, while training frequency varied significantly. It is interesting that this type of training had positive effects on elements of performance, but the optimum stimulus or the minimum training stimulus is unclear.
  3. In some studies, the training programs involved upper extremity muscle training. Yet, the discussion focuses on lower extremity tests. Is this a matter of a few studies on upper extremities or that such training in upper extremities is insufficient?
  4. Finally, this study would be really more useful, had it compared resistance band training with other training forms.

Line 220: From a mechanical point of view, power is not the ability to exert maximal force in a minimal amount of time (this is impulse or work), but is the force times velocity. So, perhaps the authors can reword this.

Author Response

Dear Reviewer,
Thank you very much for taking the time to review this manuscript. You have contributed to the
substantial improvement of our manuscript.
Please find the detailed responses and the corresponding revisions/corrections highlighted in the manuscript, attached as two separate Word files.

Kind regards,

the authors

Reviewer 2 Report

Comments and Suggestions for Authors

the topic of the paper is relevant; among other aspects, elastic bands constitute an inexpensive and easily transportable resource that can be used by athletes in team sports.

On the other hand, the application of the PRISMA protocol ensures that the systematic review meets standard requirements.

The statistical analysis is also examined through various graphs, with the forest plot standing out and being well interpreted. As a result, the findings are consistent.

The effect sizes are important and relevant; in any case, the authors acknowledge the limitations.

In summary: in my opinion, it is a good contribution to the body of evidence on elastic bands, and it also has immediate application for sports performance professionals: coaches and strength and conditioning coaches.

Regarding the English, it needs to be improved. Correct the typos.

Author Response

(The authors gave the same response as above.)

Reviewer 3 Report

Comments and Suggestions for Authors

Dear Editor,

Thank you for the opportunity to evaluate the manuscript on elastic band training. The manuscript addresses a topic pertinent to the scope of JFMK by examining the effects of resistance training with elastic bands on performance indicators in team sports, aligning with the journal’s emphasis on performance, rehabilitation, and sports therapy. The synthesis is current, including studies up to 2024, and the applied focus will interest strength and conditioning coaches, sport coaches, and physiotherapists. The reported effect sizes are substantial and, when interpreted appropriately, suggest potential practical relevance, particularly for sprint performance, change of direction, and lower-limb power. However, the abstract does not clarify whether a control group was present across all included studies, nor the nature of the comparisons performed. Because the authors present elastic bands as an alternative to traditional resistance training, it is essential, starting in the abstract, to specify the comparators employed (elastic-band training versus conventional resistance training, versus no intervention, or versus other forms of physical preparation), thereby avoiding overinterpretation. In the introduction, I recommend deepening the state of the art regarding the mechanisms by which elastic-band training may directly influence the performance variables under analysis, situating prior evidence and the gaps that justify the review.

Regarding methods, the manuscript declares PRISMA adherence and includes a flow diagram, alongside strengths such as prior registration in INPLASY, blinded screening in Rayyan, duplicate selection and data extraction, and the use of a tool to digitize graphical data when needed. Even so, there are methodological weaknesses that heighten the risk of bias: the search was restricted to PubMed and Web of Science without inclusion of important databases such as Scopus, Embase, SPORTDiscus, or regional databases. In addition, the explicit exclusion of non-English articles introduces potential location/language bias. The search strategy is generic, lacking field-specific controlled vocabulary (e.g., MeSH/tiab) and a defined time window. Risk-of-bias assessment also appears heterogeneous with respect to study design. Although the authors include randomized trials, quasi-experimental studies, and cohorts, they report using RoB 2 for “randomized studies” without clarifying which instrument was applied to non-randomized designs; in this context, ROBINS-I would be expected. The PICO eligibility is, in general, appropriate for population, intervention, comparators, and outcomes (strength, power, sprint, change of direction, and jump). The statistical synthesis, using a random-effects model and standardized mean differences (SMD), is suitable, including the correct interpretation of negative SMDs when decreases in time indicate improvement.

Nevertheless, technical gaps remain that undermine inferential robustness. Pooling different designs (trials, quasi-experiments, and cohorts) into a single effect set without subgroup analyses or estimand-specific modeling tends to inflate heterogeneity and complicates causal interpretation. Although the limitations section acknowledges substantial heterogeneity, sensitivity analyses stratified by design, methodological quality, and intervention characteristics are lacking. In high-quality reviews in this field, it is common to extend the search to three or more databases; accordingly, inclusion of SPORTDiscus and/or Scopus, registration in PROSPERO (in addition to INPLASY), and the use of design-appropriate tools for non-randomized studies are warranted. Addressing these aspects would bring the manuscript closer to the methodological standards of leading reviews. Another critical issue is that, in several included studies, the control groups received no intervention, whereas the experimental groups trained specifically the physical capacities being evaluated. This exposure asymmetry suggests that part of the observed gains may primarily reflect a “training versus no training” effect rather than an inherent superiority of elastic bands over conventional resistance methods. This distinction must be made explicit when interpreting the results.

In the discussion, the authors state that elastic-band training yields statistically significant and practically meaningful improvements in power, change of direction, and sprint, but they do not make clear that many experimental designs compared a structured intervention with the absence of intervention in the control condition. This information is crucial to qualify the scope of the conclusions. The discussion also reads too close to the results, with limited critical engagement with the existing literature and insufficient consideration of mechanisms and external validity. I recommend rewriting the section to integrate comparisons with prior studies of traditional resistance training and with related meta-analyses, discussing consistencies and discrepancies, and addressing practical applicability considering context, duration, frequency, and load prescription for elastic-band training.

For overall improvement, I suggest explicitly stating in the abstract and methods the comparators used and the rationale for how they are handled analytically; detailing and expanding the search strategy (including field-specific terms, additional databases, and temporal criteria); justifying the choice of risk-of-bias tools according to study design; conducting subgroup analyses by design and quality, as well as sensitivity analyses to assess the influence of studies with no-intervention controls; and tempering the conclusions to reflect the nature of the comparators, avoiding extrapolation to “an alternative to resistance training” when the predominant evidence derives from comparisons with no intervention. Finally, I recommend substantially expanding the discussion with stronger theoretical and contextual integration of the findings and making the limitations explicit to guide future research and to support critical appraisal by practitioners. Considering the above, I recommend major revisions: although the topic fit is strong and the aggregated effects are promising, methodological and transparency gaps should be addressed prior to consideration for acceptance.

Sincerely,

Reviewer.

Author Response

(The authors gave the same response as above.)

Round 2

Reviewer 3 Report

Comments and Suggestions for Authors

Dear Authors,

I appreciate the revised manuscript that was sent. After reading the modifications, I realize that you made only superficial changes. I suggest that in the introduction, you delve deeper into the physiological mechanisms involved in training with elastic bands. You only highlighted the effects. In the Methods and Results section, I suggest following the suggestions made in the first review. I emphasize that the research you conducted is already a year out of date, and the reduction in the database search (considering only two databases) limits the innovative power of the research, as well as the impact of the data. I believe that after expanding the research, it will be easier to proceed with the suggestions made for the results. If the content of the results does not change profoundly, I believe the discussion will need few adjustments, since it is well-constructed. Unfortunately, it is not possible to consider publication without considering the methods and results. Therefore, I suggest major corrections.

Best regards,

Reviewer

Author Response

Dear Reviewer,

Thank you very much for taking the time to review this manuscript. You have contributed to its substantial improvement.

Please find the detailed responses and the corresponding revisions/corrections highlighted in the attached documents on the electronic platform.

Kind regards,

the authors

Round 3

Reviewer 3 Report

Comments and Suggestions for Authors

Dear Editor, I appreciate the opportunity to review the manuscript again. Given the changes made, the manuscript should be accepted.

Sincerely,

Reviewer